# Polyphenol Oxidase as a Promising Alternative Therapeutic Agent for Cancer Therapy

**DOI:** 10.3390/molecules27051515

**Published:** 2022-02-23

**Authors:** Qinqin Yuan, Huixia Guo, Jiajie Ding, Chan Jiao, Yalei Qi, Hajra Zafar, Xueyun Ma, Faisal Raza, Jianqiu Han

**Affiliations:** 1College of Ecological Technology and Engineering, Shanghai Institute of Technology, Shanghai 201418, China; qinqin_yuan@163.com (Q.Y.); guohuixia0424@163.com (H.G.); duwnyu@hotmail.com (J.D.); 2Shanghai Key Laboratory of Regulatory Biology, Institute of Biomedical Sciences, School of Life Sciences, East China Normal University, Shanghai 200241, China; jiaochan1201@163.com (C.J.); ylqi@bio.ecnu.edu.cn (Y.Q.); 3School of Pharmacy, Shanghai Jiao Tong University, Shanghai 200240, China; hajrazafar@sjtu.edu.cn

**Keywords:** PPO, cancer cells, proliferation, migration, invasion, apoptosis

## Abstract

Cancers have always been the most difficult to fight, the treatment of cancer is still not considered. Thus, exploring new anticancer drugs is still imminent. Traditional Chinese medicine has played an important role in the treatment of cancer. Polyphenol oxidase (PPO) extracted from *Edible mushroom* has many related reports on its characteristics, but its role in cancer treatment is still unclear. This study aims to investigate the effects of PPO extracted from *Edible mushroom* on the proliferation, migration, invasion, and apoptosis of cancer cells in vitro and explore the therapeutic effects of PPO on tumors in vivo. A cell counting kit-8 (CCK8) assay was used to detect the effect of PPO on the proliferation of cancer cells. The effect of PPO on cancer cell migration ability was detected by scratch test. The effect of PPO on the invasion ability of cancer cells was detected by a transwell assay. The effect of PPO on the apoptosis of cancer cells was detected by flow cytometry. Female BALB/c mice (18–25 g, 6–8 weeks) were used for in vivo experiments. The experiments were divided into control group, model group, low-dose group (25 mg/kg), and high-dose group (50 mg/kg). In vitro, PPO extracted from *Edible mushroom* significantly inhibited the proliferation, migration, and invasion capability of breast cancer cell 4T1, lung cancer cell A549, and prostate cancer cell C4-2, and significantly promoted the apoptosis of 4T1, A549, and C4-2. In vivo experiments showed PPO inhibitory effect on tumor growth. Collectively, the edible fungus extract PPO could play an effective role in treating various cancers, and it may potentially be a promising agent for treating cancers.

## 1. Introduction

Cancers have always been a malignant disease with high mortality in diseases. In recent decades, with an increasing number of patients with cancer in social, economic, and electronic technology, the upward trend is becoming more and more severe [1]. There are many studies on the treatment of cancer, for example, some synthetics and their derivatives can play an anti-tumor effect on breast cancer [2,3]; at the same time, the synthesis of small molecules [4,5,6], cell targeting [7] and drug targeting [8] have also become novel paths for the treatment of breast cancer. Androgen deprivation therapy, targeted androgen therapy, immunotherapy, and other treatment methods [9,10,11,12] are currently the most effective treatment for prostate cancer. In addition, T cells by regulate the mitochondrial activity of ovarian cancer cells [13]. There are also studies that use synthetic lethal protocols to defeat KRAS mutant carcinoma by targeting the activation of tumor suppressors [14] to treat lung cancer. Still, its therapeutic effect is not yet clear. At present, there are many chemotherapy drugs on the market, such as temozolomide, doxorubicin, and cisplatin. However, the side effects of these drugs on the human body are becoming increasingly obvious and cannot be ignored. Doxorubicin can produce serious cardiac toxicity [15], and temozolomide can cause blood, skin, eyes, and gastrointestinal reactions [16]. Therefore, it is still urgent to explore new drugs for cancer treatment.

Traditional Chinese medicine (TCM) has a history of thousands of years and is widely used in oncology [16]. About 30% of small molecules approved by the Food and Drug Administration between 1999 and 2008 are compounds or derivatives, both from natural products [17]. A major obstacle to tumor chemotherapy is the development of multidrug resistance (MDR). Continuous use of chemotherapeutic drugs during tumor treatment can induce the expression of ATP-dependent transporter P-Glycoprotein (P-gp, MDR1), thereby reducing the accumulation of chemotherapeutic drugs in cells and leading to MDR [18]. TCM is a treasure in the development of tumor-related drugs. A large number of potential compounds derived from traditional Chinese medicine have been confirmed to reverse multidrug resistance. For example, flavonoids can interact with P-gp through the planar structure of flavonoids and the hydrophobic amino acid of P-gp, which inhibits P-pg and increases the toxicity of chemotherapy drugs [17]. Doxorubicin (DOX) is one of the most widely used cytotoxic drugs for the treatment of breast cancer. Quercetin (QUE) is a typical flavonoid drug, which is widely distributed in vegetables and fruits. QUE can increase the cytotoxicity of DOX by downregulating the expression of P-gp, while biotin can increase cell uptake through receptor-mediated endocytosis, thereby overcoming MDR [15]. Furthermore, there are some traditional Chinese medicines used in the ranks of anticancer drugs, such as crocin which induces apoptosis of breast cancer cells by increasing ROS production in the cytoplasm and mitochondria [19]. Polyphenols, alkaloids, and polysaccharides have anti-tumor effects by regulating death receptor-mediated apoptosis and mitochondrial-dependent cell death [17]; *Solanum nigrum* L. methanol extract inhibits the proliferation, migration, and invasion of C6 high-grade gliomas cells, induces their apoptosis, and hinders the growth and infiltration of C6 gliomas in the brain [16]. In addition, Liu-jun-zi-tang, Coumarin, Aescine, and other Chinese medicinal materials can improve postoperative fatigue, anorexia, diarrhea, nausea, vomiting, pain, lymphedema, and other symptoms; Huang-Qin, Ginseng, BanZhiLian, Huachansu injection, Shenqi fuzheng injection, and Kanglaite injection combined with chemotherapy or radiotherapy can improve the curative effect, reduce the side effects and complications caused by chemotherapy and radiotherapy on the human body [20]. Folic acid is ubiquitous in our lives, mostly in our food, and it is also a unique component in traditional Chinese medicine. A new Hal-based drug delivery system aims to improve the bioavailability and biological activity of doxorubicin in cancer treatment. This nano-delivery system can target HeLa cells through folic acid receptor-mediated endocytosis, inhibit tumor cell proliferation, and even induce cell death, but its cytotoxicity to normal cells is relatively low [21]. *Saponaria prostrata* WILLD. subsp. *anatolica* HEDGE significantly inhibited cell colony formation and wound closure in breast cancer and colorectal cancer, inhibited angiogenesis, and triggered apoptosis [22]. In short, traditional Chinese medicine plays an important role in adjuvant tumor treatment.

Over the past few decades, more and more studies have shown that medicinal fungi have the potential to fight cancer [23,24]. *Sanghuangprous vaninii* is a medicinal fungus. It has been reported that *Sanghuangprous vaninii* has anti-tumor activity against cancer. For example, blocking cell cycle at G0/G1 phase in B16F10 melanoma cells [25], inducing apoptosis and ROS generation in colon cancer cell HT-29 [26], and inhibiting the proliferation of MDA-MB-231 breast cancer cells [27]. The extract of *Sanghuangprous vaninii* inhibits cervical cancer by blocking cell cycle and inducing apoptosis [28]. *Edible mushroom* is one of many medicinal fungi. With the improvement of people’s quality of life, *Edible mushroom* have become one of the essential foods on the table.

PPO is a kind of protein, which is widely distributed in the plastid of plants, fungi, and insects [29,30,31]. It catalyzes the *o*-hydroxylation of monophenols to *o*-diol (tyrosinase activity, Enzyme Committee 1.14.18.1) and the oxidation of *o*-diphenols to quinones (catecholase activity, Enzyme Committee 1.10.3.2) [30,31,32,33,34]. At the same time, because it can catalyze the oxidation of phenol or aromatic amines, it is also widely used in various fields such as medicine and health. It has also become the direction of future drug development efforts to treat Parkinson’s disease and Alzheimer’s. PPO is also one of the earliest studied enzymes due to its convenient detection. At present, there are many reports on plant and fungal PPO, such as the activity of blueberry PPO [35,36], the antioxidant activity of mussel PPO [37,38], the characteristics of kiwifruit PPO [36,39], and the characteristics of plum PPO [38,40]. However, the cancer resistance of PPO is rarely reported. In this study, the cancer resistance of PPO was studied.

The PPO used in this study is from *Edible mushroom*, the enzyme activity is 1024 u/mg, molecular weight is 128 kDa, soluble in water. It was used for in vitro effects on proliferation, migration, invasion, and apoptosis of breast cancer cells, ovarian cancer cells, prostate cancer cells, and lung cancer cells, and in vivo experiments were carried out based on in vitro experiments.

## 2. Results

### 2.1. PPO Inhibited Proliferation and Survival of Cancer Cells

We first analyzed the effects of PPO on the viability of breast cancer cells, ovarian cancer cells, prostate cancer cells, and lung cancer cells. Fitting the dose-response basis function can be used to estimate the half-maximal inhibitory concentration (IC_50_) [16]. Different concentrations of PPO were added to the cell culture medium, and the cells were cultured for 24 h. Endpoint determinations were made with CCK8, in which light absorption was measured at 450 nm by ELISA (enzyme-linked immunoassay) to reflect the number of living cells. The results for each cell line are reported as shown in Figure 1, among them are the normal cells of humans and mice which show a relatively high cell viability after 24 h of incubation. All the cancer cells were inhibited at different degrees, respectively.

Even at a high concentration of 0.5 mg/mL, the viability of GES-1 and 3T3 remained above 80%. In contrast, all the kinds of cancer cells’ performance decreased viability in a dose-dependent manner. Among the three lines of breast cancer cells, the mice breast cancer cell 4T1 was most significantly decreased, the human breast cancer cell T-47D was significantly decreased, and MCF-7 was suppressed to a certain extent. For the five lines of ovarian cancer cells: A2780 was most significantly decreased, while the other four lines of cancers of Hey, OVCAR-8, OVISE, and SK-OV-3 were suppressed to a certain degree. The viability of two lines of prostate cancer cells, C4-2 and DU 145, exhibited significant depression. The inhibitory effect of C4-2 was the most significant. For the two lines of lung cancer A549 and PC9, the inhibitory effect of A549 was significant, while the PC9 was not significant (Table 1). With the increase of PPO concentration, the viability of cancers cells showed a downward trend.

### 2.2. PPO Inhibited Cancer Cells Invasion

Invasion is a key factor for cancer cell expansion and metastasis. We examined whether the PPO modulates cancer cells’ invasion through the transwell chamber coated with Matrigel. Based on the CCK8 assay, we chose a human gastric normal cell of GES-1, a breast cell of 4T1, a prostate cancer cell of C4-2, and a lung cancer cell of A549 for the invasion assay. The results showed that when PPO concentration was 0.5 mg/mL, the inhibitory effect on the invasion of normal gastric epithelial cell GES-1 was obvious (Figure 2A). However, for the three kinds of cancer cells, PPO at 0.005 mg/mL concentration has shown a significant inhibitory effect on cell invasion. Among them, PPO had the most obvious inhibitory effect on C4-2, and the invasion rate was almost zero (Figure 2B). The lung cancer cell of A549 and the breast cancer cell of 4T1 were all significantly inhibited by the PPO at the concentration was 0.005 mg/mL, the invasion rate of these cancer lines were reduced to 50% (Figure 2C,D). It showed that PPO could effectively inhibit cancer cells’ invasion mobility of the prostate/lung/breast cancer cells of C4-2/A549/4T1 and the normal gastric epithelial cell GES-1. Still, the inhibitory effect on GES-1 cells was significantly lower than that of the above three cancer cells (Figure 2E–G).

### 2.3. PPO Inhibited Cancer Cell Migration

To determine the inhibitory effect of PPO on cancer migration, the wound healing migration assay was utilized. Based on the invasion assay, the cancer cell lines of A549, C4-2, 4T1, and human gastric normal cell GES-1 were selected. After exposing these cells to different PPO concentrations, wound closures were monitored and photographed to count the migrating cells. In the wound healing assay, the most significant inhibitory effect of PPO on C4-2 cells and 4T1 cells after 24 h were monitored with the concentration of 0.005 mg/mL (Figure 3B,D). A549 cells showed a significantly inhibited effect after 24 h with the concentration of 0.05 mg/mL, and the inhibition rate reached about 30% (Figure 3C). When the concentration of PPO was 0.05 mg/mL, the inhibitory effect on GES-1 migration of normal gastric epithelial cells was obvious, but the inhibitory rate was about 80% (Figure 3A). This suggested that PPO can effectively inhibit the migration of prostate/lung/breast cancer cells of C4-2/A549/4T1 and normal gastric epithelial cell GES-1. Still, the inhibitory effect on GES-1 cells was significantly lower than that of the above three cancer cells (Figure 3E–G).

### 2.4. Effect of PPO on Cancer Cell Apoptosis

Human gastric normal cell GES-1, prostate cancer cell C4-2, lung cancer cell A549, and breast cancer cell 4T1 were selected for this assay. After these cells were exposed to different concentrations of PPO for 24 h, the apoptosis of cells was evaluated by flow cytometry. The results showed that PPO inhibited the apoptosis of normal gastric epithelial cells GES-1 (Figure 4A). However, it showed that the apoptosis of the three kinds of cancer cells increased with PPO concentration. The apoptosis of the three cancer cells was significant at the concentration of 0.05 mg/mL. At the concentration of 0.5 mg/mL, the apoptosis rate of the three cell lines exceeded 50%, and they were more inclined to apoptosis in the later stage. However, compared with A549, 4T1 and C4-2 showed a little more (Figure 4B–D). The result showed that PPO could significantly promote the apoptosis of lung cancer cells of A549, prostate cancer cells of C4-2, and breast cancer cell 4T1, but inhibiting GES-1 apoptosis (Figure 4E–G).

### 2.5. Drug Evaluation In Vivo

Through tracking data, PPO had a significant inhibitory effect on the growth of breast cancer (Figure 5B). Additionally, over the course of treatment, the effect of PPO on the weight of the mice was less than the effect of the current anticancer drug cisplatin on the weight of the mice (Figure 5A). After the experiment, the mice tumors were dissected, photographed, and weighed (Figure 5D). Finally, it was found that PPO also had a certain inhibitory effect on the weight of breast cancer cells in the body (Figure 5C). In addition, it was found in the experiment that the inhibitory effect of PPO on 4T1 was dose dependent.

## 3. Discussion

Breast cancer is one of the malignant diseases and one of the biggest enemies of women, with the title of the “pink killer” [41,42,43]. Prostate cancer is second only to lung cancer in men [8,9,44]. The prevalence of ovarian cancer should not be underestimated for Chinese women [45]. Lung cancer ranks first in the global cancer mortality rate for several consecutive years and is the main cause of cancer-related deaths worldwide [46]. Traditional Chinese medicine has a long history and plays an important role in treating human diseases. PPO is a dinuclear copper-centered enzyme, widely distributed in plants, fungi, microorganisms, and animals [47] as an antioxidant enzyme that can oxidize high phenol and non-phenol aromatic compounds [48]. In addition, it is involved in oxygen scavenging, pigment formation, and defense mechanisms against plant pathogens and herbivores [38]. This study described a new anticancer drug PPO extracted from *Edible mushroom*, which showed good cancer resistance to breast cancer, prostate cancer, lung cancer, and ovarian cancer.

It is well recognized that the cancer cell viability is assumed generally evaluated by CCK8 assay [5,39,40,41,42]. In this study, the IC_50_ was used to preliminarily evaluate the PPO’s anti-proliferative effect of breast, ovarian, prostate, and lung cancer cells. According to the results, it can be seen that PPO has different inhibitory effects on different types of cancer cells and different cell lines of the same type of cancer. The growth and metastasis characteristics of 4T1 cells in BALB/c mice are very similar to those in human breast cancer, which is an animal model of human VI stage breast cancer. MCF-7 cells retain the characteristics of multiple differentiated mammary epithelial cells, which contain Tx-4 oncogene. T-47D cell-expressed WNT7B oncogene. DU 145, C4-2 cells were taken from different parts of prostate cancer patients, and the expression of prostate antibodies was different. PC9 cells are erlotinib-resistant human lung adenocarcinoma cells. A549 cells can synthesize a high proportion of unsaturated fatty acid lecithin through the cytidine diphosphate choline pathway. A2780, SK-OV-3, OVISE, OVCAR-8, Hey cells were taken from different parts of ovarian cancer patients, and their tolerance to tumor necrosis factor and several cytotoxic drugs, including cisplatin and doxorubicin, was different. Drugs can specifically bind to specific receptors on the surface of cancer cells for endocytosis, thereby effectively killing tumor cells [49]. Therefore, PPO obtained in the CCK8 experiment showed different inhibitory effects on the activity of different cancer cells of the same cancer, which may be because these cancer cells came from different parts of different patients and formed different specific receptors. PPO could specifically bind to their different specific receptors and have different degrees of endocytosis, thus effectively killing tumor cells. However, the mechanism of PPO on their cell viability needs further study.

Invasion is closely related to cancer therapeutic efficacy. Therefore, inhibition of invasion can contribute to cancer treatment. In the present study, the effect of PPO on invasion was determined. The results showed that PPO could effectively inhibit the invasion of prostate cancer cell C4-2, lung cancer cell A549 and breast cancer cell 4T1. The invasion of cancer cells mediates its metastasis and infiltration into normal tissues, which determine the degree of malignancy and the difficulty of treatment [50]. As a medium of cell invasion, the MMP family is considered as a potential tumor progression marker. They allow tumor cells to invade through the extracellular matrix and vascular basement membrane [51]. We speculated that PPO downregulated the expression of the MMP family was the main reason PPO inhibited the cancer cells’ invasion.

According to the results, PPO can effectively inhibit the migration of prostate cancer cell C4-2, lung cancer cell A549, and breast cancer cell 4T1. Two major MAP kinases, extracellular signal-regulated kinase (ERK) and c-Jun N-terminal kinase (JNK), regulate different cellular activities, even if they share some common substrates, such as c-jun, which is an inducible transcription factor that guides gene expression in response to a variety of extracellular stimuli [52]. ERK participates in various functions of angiogenesis, including cell proliferation, migration, and survival [53]. We speculate that PPO inhibits the activation of ERK and then inhibits the migration of cancer cells.

The results showed that PPO could effectively promote the apoptosis of prostate cancer cell C4-2, lung cancer cell A549, and breast cancer cell 4T1. JNK mediates signal transduction from cell surface receptors to the nucleus and regulates some anti-apoptotic proteins by regulating programmed cell death [54]. The ER is a key organelle for protein synthesis and Ca^2+^ storage [55]. When subjected to unfolded protein accumulation, ER stress is a self-protective response of cells, called UPR. Cell death response via mitochondrial pathway under excessive and irreparable toxic stress [56]. CHOP, a transcription factor, is activated by ER stress [57]. CHOP activation induces apoptosis [58]. CHOP downregulates Bcl-2 expression and upregulates Bax expression under severe ER stress [59]. The mitochondrial apoptosis pathway is a classical form of cell death, strictly regulated by Bcl-2 family proteins [60]. Therefore, we can speculate that PPO may induce apoptosis of cancer cells by inhibiting the expression of BCL-2 protein and promoting the expression of Bax.

Cell proliferation is crucial for cancer cell metastasis. The results showed that PPO had a significant inhibitory effect on the proliferation of C4-2, A549, and 4T1 cancer cells but a stronger inhibitory effect on the invasion and migration of cancer cells, indicating that anti-invasion may be the main contributor to its anti-metastasis activity.

Therefore, the PPO exhibited anticancer activity in all cancer types used in this study. The current result suggested that PPO had great potential anti-tumor and antioxidant agents. In general, the beneficial effect of PPO may be attributable to phytochemicals including antioxidants, counteract free radicals by donating protons to free radicals, and terminate potentially damaging chain reactions. The study suggested that PPO in the medium also performed antioxidant functions by reducing superoxide through the formation of hydroperoxide intermediates, but further exploration is needed to verify this.

## 4. Materials and Methods

### 4.1. Experimental Materials

Polyphenol oxidase (PPO) extracted from *Edible mushroom* was purchased from HeFei BoMei Biotechnology Co., Ltd. (BoMei, Hefei, China). The cell lines were obtained from the Medical Institute of East China Normal University. The normal cells of GSE-1 and 3T3, breast cancer cells of MCF-7, T-47D, 4T1, and ovarian cancer cells of OVISE, A2780, Hey, SK-OV-3, OVCAR-8 were cultured in DMEM (Gibco, USA) medium at 37 °C and 5% CO_2_. Lung cancer cells of A549 and PC9 and prostate cancer cells of C4-2 and DU 145 were cultured in RPMI 1640 (Gibco, USA) medium at 37 °C and 5% CO_2_. The mediums contained penicillin and streptomycin (Gibco, USA) and were supplemented with 10% FBS (fetal bovine serum, Gibco, USA). Cisplatin and BALB/c mice were obtained from the Experimental Animal Center of East China Normal University.

### 4.2. Experimental Methods

#### 4.2.1. Measurement of Cell Viability

A CCK8 assay was used to determine the viability of cancer cells [16]. Cells were seeded at 8 * 10^3^/well at the logarithmic growth phase in a 96-well plate. Cells were treated with PPO at the concentration of 0, 0.005, 0.05, and 0.5 mg/mL. After 24 h, cell counting kit-8 (CCK8) (BD, USA) was used to measure the cell viability.

#### 4.2.2. Cell Invasion Assay

A Matrigel™ invasion chamber (Corning, USA) containing 8.0 µm-pore polycarbonate membrane was put into a 24-well plate. Cells were seeded at a density of 1 * 10^4^ cells/well in 400 µL FBS-free cell culture medium containing the specified concentration of PPO in the upper chamber of Matrigel™, and 500 µL cell culture medium with FBS containing the same PPO concentration as the upper chamber was added to the lower chamber. After 24 h, use a cotton swab to remove uninvaded cells in the upper chamber, fix with 4% paraformaldehyde, stained with 2% crystal violet staining solution, and count with a light microscope.

#### 4.2.3. Cell Migration Assay

The cells were seeded in a 6-well plate for the wound healing assay. When the cells grow to >90% confluence. The scratch was performed in the well center with pipette tips and washed with PBS. Subsequently, the cells were cultured in their mediums with different concentrations of PPO for 24 h. The cells moved into the wound area were monitored and photographed. The distance between the leading edge of the migrating cells and the wound edge was monitored.

#### 4.2.4. Cell Apoptotic Assay

The cells were seeded in a 6-well plate with different concentrations of PPO for 24 h. After washing and centrifugation, quantitative measurement of apoptotic cells was conducted by dual staining with Annexin V-FITC/PI, as described previously [61]. Cells were stained with Annexin V-FITC staining apoptosis detection kit (BD, USA) at room temperature for 15 min. Then the samples were sent to detect by flow cytometry.

#### 4.2.5. Animal Model Construction

Based on experiment 3.2.1, the 4T1 cell was selected for in vivo drug evaluation. 4T1 cells were collected and injected under BALB/c mice’s left rib when they were cultured to the appropriate density. Animal studies were performed according to the guidelines approved by the Ethical Review of Laboratory Animals of East China Normal University. When the tumor lengths reached 80–100 mm^3^, they were divided into groups. The experimental groups were divided into control group, model group, low-dose group (25 mg/kg), and high-dose group (50 mg/kg), with five in each group.

#### 4.2.6. Statistical Analysis

In this study, the experimental and control groups were analyzed by a *t*-test, and the data were presented by standard deviation and average value (s, d). GraphPad Prism 6.0 software was used for statistical analysis in the data analysis process. A *p* value ≤ 0.05 was considered statistically significant.

## 5. Conclusions

The anticancer properties of *Edible mushroom* extract PPO were investigated. The results indicated that it had a strong inhibition of all selected cell types and had a strong inhibitory effect on 4T1 and C4-2 cells with inhibition percentages of 50.7–66.9%. The results support that PPO may be a protecting agent for cancer treatment. However, the mechanism of PPO on cancer cells needs further study.

## Figures and Tables

**Figure 1 molecules-27-01515-f001:**
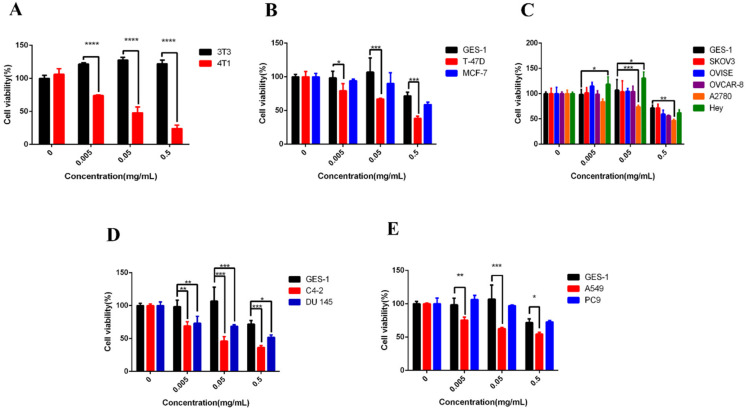
Effect of polyphenol oxidase on (**A**) breast cancer cell 4T1 and mice normal fibroblast 3T3, (**B**) human breast cancer cells MCF-7, T-47D and human gastric normal cell GES-1, (**C**) human gastric normal cell GES-1 and human ovarian cancer cells A2780, Hey, OVISE, OVCAR-8, SK-OV-3, (**D**) human gastric normal cell GES-1 and human prostate cancer cells C4-2, DU 145, (**E**) human gastric normal cell GES-1 and human lung cancer cells A549, PC9; bars, SE (* *p* < 0.05; ** *p* < 0.01; *** *p* < 0.001; **** *p* < 0.0001 versus control).

**Figure 2 molecules-27-01515-f002:**
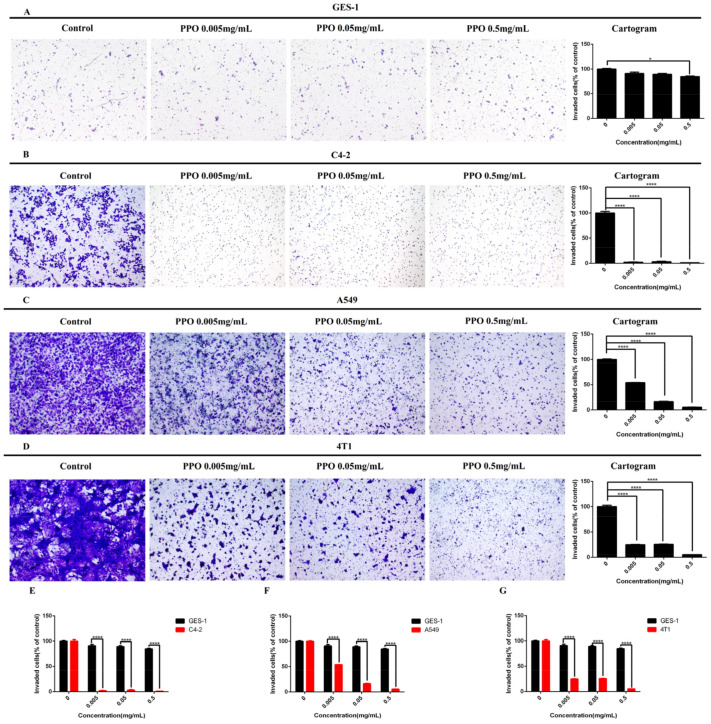
Effect of PPO on the invasion of (**A**) human gastric normal cell GES-1, (**B**) prostate cancer cell C4-2, (**C**) lung cancer cell A549, (**D**) breast cancer cell 4T1. Comparison of the effects of PPO on the invasion of (**E**) prostate cancer cell C-2, (**F**) lung cancer cell A549, (**G**) breast cancer cell 4T1 and human gastric normal cell GES-1; bars, SE (* *p* < 0.05; **** *p* < 0.0001 versus control).

**Figure 3 molecules-27-01515-f003:**
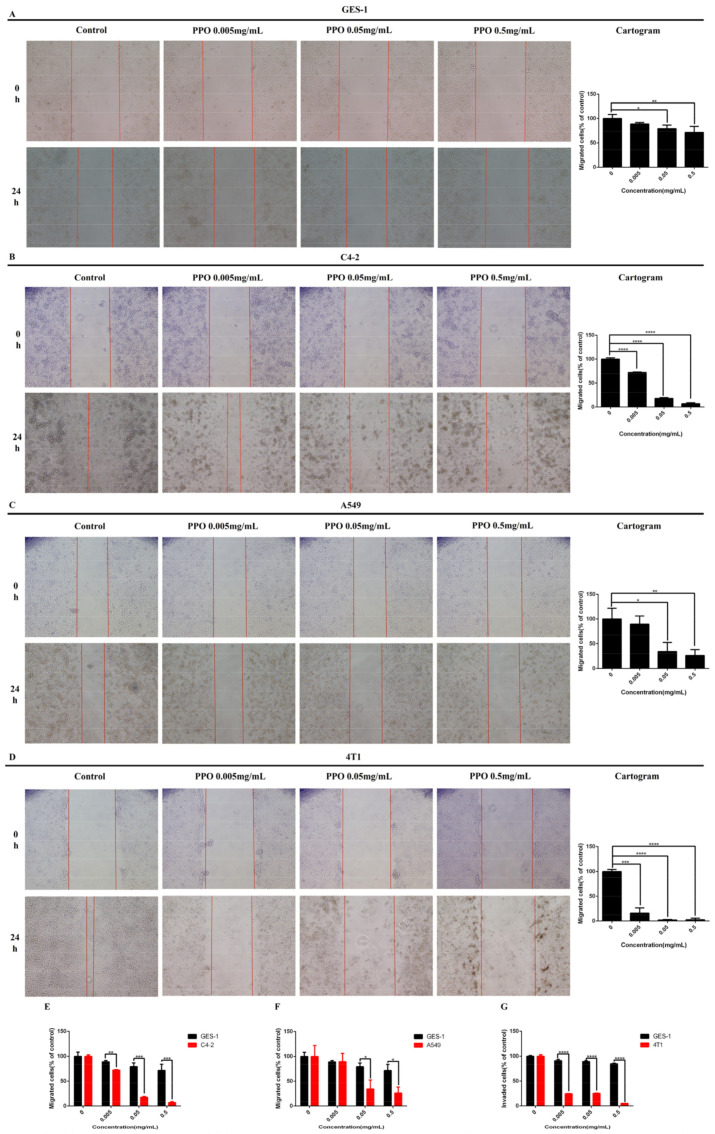
Effect of PPO on the migration of (**A**) human gastric normal cell GES-1, (**B**) prostate cancer cell C4-2, (**C**) lung cancer cell A549, (**D**) breast cancer cell 4T1. Comparison of the effects of PPO on the migration of (**E**) prostate cancer cell C-2, (**F**) lung cancer cell A549, (**G**) breast cancer cell 4T1 and human gastric normal cell GES-1; bars, SE (* *p* < 0.05; ** *p* < 0.01; *** *p* < 0.001; **** *p* < 0.0001 versus control).

**Figure 4 molecules-27-01515-f004:**
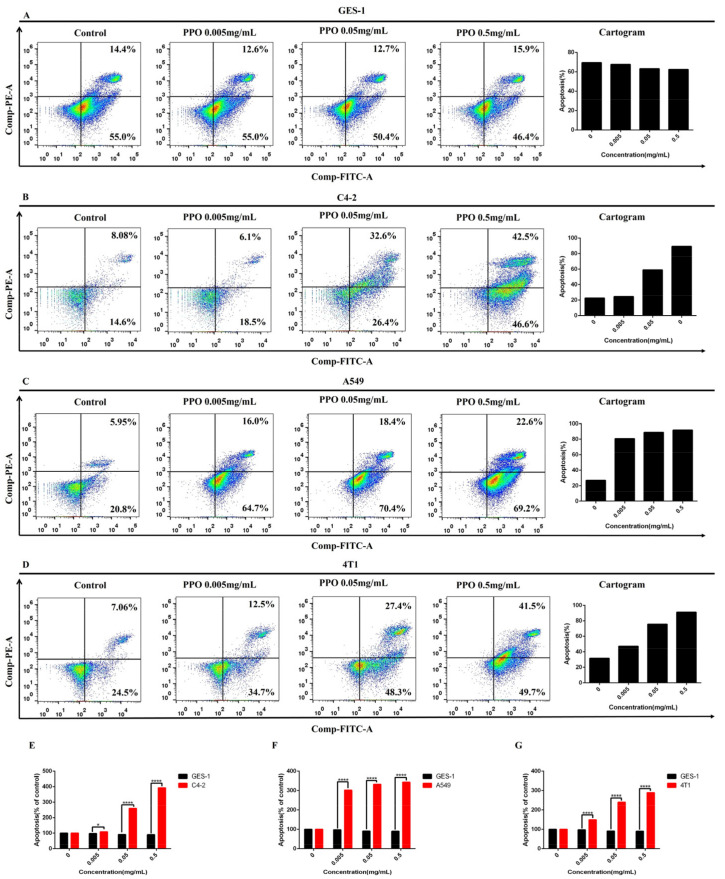
Effect of PPO on the apoptosis of (**A**) human gastric normal cell GES-1, (**B**) prostate cancer cell C4-2, (**C**) lung cancer cell A549, (**D**) breast cancer cell 4T1. Comparison of the effects of PPO on the apoptosis of (**E**) prostate cancer cell C-2, (**F**) lung cancer cell A549, (**G**) breast cancer cell 4T1 and human gastric normal cell GES-1; bars, SE (* *p* < 0.05; **** *p* < 0.0001 versus control).

**Figure 5 molecules-27-01515-f005:**
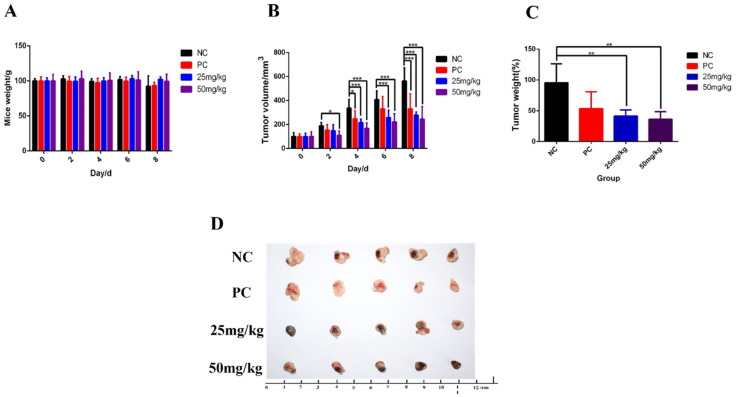
Effects of PPO on (**A**) body weight of mice, (**B**) tumor volume, (**C**) tumor weight. (**D**) Tumor anatomy of mice in each group; bars, SE (* *p* < 0.05; ** *p* < 0.01; *** *p* < 0.001 versus control).

**Table 1 molecules-27-01515-t001:** Effects of PPO on cell viability of 3T3 ^1^, 4T1 ^2^, GES-1 ^3^, T-47D ^4^, MCF-7 ^5^, A2780 ^6^, SK-OV-3 ^7^, OVISE ^8^, OVCAR-8 ^9^, Hey ^10^, C4-2 ^11^, DU 145 ^12^, A549 ^13^, PC9 ^14^.

Cell Name	IC_50_/mg/mL
3T3	0.928
4T1	0.047
GES-1	0.492
T-47D	0.097
MCF-7	0.167
A2780	0.217
SK-OV-3	0.368
OVISE	0.237
OVCAR-8	0.461
Hey	0.460
C4-2	0.026
DU 145	0.175
A549	0.083
PC9	0.103

^1^ 3T3 is a mouse embryonic fibroblast. ^2^ 4T1 is a mouse breast cancer cell. ^3^ GES-1 is a human gastric normal cell. ^4^ T-47D is a human mammary duct carcinoma cell line. ^5^ MCF-7 is a human breast cancer cell. ^6^ A2780 is a human ovarian cancer cell. ^7^ SK-OV-3 is a human ovarian cancer cell. ^8^ OVISE is a human ovarian cancer cell. ^9^ OVCAR-8 is a human ovarian cancer cell line. ^10^ Hey is a human ovarian cancer cell. ^11^ C4-2 is human prostate cancer cell. ^12^ DU 145 is a human prostate cancer cell. ^13^ A549 is a human non-small cell lung cancer cell line. ^14^ PC9 is a human non-small cell lung cancer cell line.

## Data Availability

The authors declare that the data supporting this study are available within the paper. All other data are available from the authors upon reasonable request.

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
