# Peer review of "Polyphenol Oxidase as a Promising Alternative Therapeutic Agent for Cancer Therapy"

_molecules, 2022, doi:10.3390/molecules27051515_

Round 1

Reviewer 1 Report

This version of the work of Yuan et all is significantly improved regarding their first submitted version. Still, it requires changes concerning:

1) Language errors (and/or lack of spaces) throughout the manuscript 2) The discussion section must come before the methods. 3) References must follow the journal's guidelines.

Author Response

Reviewer 1:

This version of the work of Yuan et all is significantly improved regarding their first submitted version. Still, it requires changes concerning:

  • Language errors (and/or lack of spaces) throughout the manuscript

Response: Thanks for your valuable comment. Following your kind suggestion, all documents have been modified (including the language errors and/or lack of spaces) in the revised manuscript.

  • The discussion section must come before the methods.

Response: Thanks for your valuable suggestion. Following your kind suggestion, the discussion part has been modified to the front of the methods in the revised manuscript.

  • References must follow the journal's guidelines.

Response: Thanks for your valuable recommendation. In the revised manuscript, the references have been updated according to the Molecules guidelines.

Reviewer 2 Report

Authors:

You have followed all my recommendations. The revised version of the manuscript has much higher scientific value than the original manuscript.

However, I have found that references are presented in a bit strange way. Mostly only the first author's name appears, followed by "et al." I have a feeling that all authors of any cited paper have the same right to be cited. I recommend to revise this part of the manuscript.

Author Response

Reviewer 2:

You have followed all my recommendations. The revised version of the manuscript has a much higher scientific value than the original manuscript.

However, I have found that references are presented in a bit strange way. Mostly only the first author's name appears, followed by "et al." I have a feeling that all authors of any cited paper have the same right to be cited. I recommend to revise this part of the manuscript.

Response: Thanks for your valuable recommendation. Following your kind suggestion, we have included all authors and follow the journal guidelines in the revised manuscript.

Reviewer 3 Report

Comments:

  1. Page 1, line 20, “the effects of PPO extracted from edible mushroom on” should be amended “the effects of PPO extracted from edible mushroom on”
  2. Page 1, line 27-28, “low-dose group (25mg/kg) and high-dose group (50mg/kg).” should be amended “low-dose group (25 mg/kg) and high-dose group (50 mg/kg).”.
  3. Page 3, line 103, “o-diphenols to quinones” should be amended “o-diphenols to quinones”.
  4. Page 5, line 143, “Fig 1 Effect” should be amended “Fig 1. Effect”.
  5. Page 6, in Fig 2, Fig 3. and Fig 4. “Cartogram”, image resolution is too poor and the text is too small, authors should increase their image resolution or figure size.
  6. Authors should list each work in the bibliography, and should also refer to the format of the bibliography in "Molecules". Ex samples: 1. Author 1, A.B.; Author 2, C.D. Title of the article. Abbreviated Journal NameYearVolume, page range.
  7. Authors should add descriptions of PPO properties in the article, such as molecular weight, source, purity content, etc.

Author Response

Reviewer 3:

  1. Page 1, line 20, “the effects of PPO extracted from edible mushroom on” should be amended “the effects of PPO extracted from Edible mushroom on”

Response: Thanks for your valuable recommendation. Following your kind suggestion, we have amended “the effects of PPO extracted from edible mushroom on” to “the effects of PPO extracted from Edible mushroom on” in the revised manuscript.

  1. Page 1, line 27-28, “low-dose group (25mg/kg) and high-dose group (50mg/kg).” should be amended “low-dose group (25 mg/kg) and high-dose group (50 mg/kg).”.

Response: Thanks for your valuable recommendation. Following your kind suggestion, we have amended “low-dose group (25mg/kg) and high-dose group (50mg/kg).” to “low-dose group (25 mg/kg) and high-dose group (50 mg/kg).” in the revised manuscript.

  1. Page 3, line 103, “o-diphenols to quinones” should be amended “o-diphenols to quinones”.

Response: Thanks for your valuable recommendation. Following your kind suggestion, we have amended “o-diphenols to quinones” to “o-diphenols to quinones” in the revised manuscript.

  1. Page 5, line 143, “Fig 1 Effect” should be amended “Fig 1. Effect”.

Response: Thanks for your valuable recommendation. Following your kind suggestion, we have amended “Fig 1 Effect” to “Fig 1. Effect” in the revised manuscript.

  1. Page 6, in Fig 2, Fig 3. and Fig 4. “Cartogram”, image resolution is too poor and the text is too small, authors should increase their image resolution or figure size.

Response: Thanks for your valuable recommendation. In the revised manuscript, we have increased the resolution of the image on page 6, in Fig 2, Fig 3 and Fig 4, and made the text larger and bolder. The figure below shows:

Fig 2:

Fig 3:

Fig 4:

  1. Authors should list each work in the bibliography, and should also refer to the format of the bibliography in "Molecules". Ex samples: 1. Author 1, A.B.; Author 2, C.D. Title of the article. Abbreviated Journal NameYear, Volume, page range.

Response: Thanks for your valuable recommendation. Following your kind suggestion, we have included all authors and follow the journal guidelines in the revised manuscript.

  1. Authors should add descriptions of PPO properties in the article, such as molecular weight, source, purity content, etc.

Response: Thanks for your valuable recommendation. Following your kind suggestion, in the revised manuscript, we have added a description of PPO properties, such as molecular weight, source, purity content, etc.

This manuscript is a resubmission of an earlier submission. The following is a list of the peer review reports and author responses from that submission.

Round 1

Reviewer 1 Report

The work of Yuan et al is of interest to be considered for publication in Molecules, but it would require extensive revisions. 

-    The manuscript does not conform the journal rules (eg Methods should be section 3 instead of 2; references not formatted...). 
-    There are many language mistakes (including the lack/extra spaces in sentences). Revise all the document.
-    Title: please change to “Polyphenol oxidase as a promising…”
-    Abstract: Do not consider a clear division of the abstract (delete background; objective….)
-    Introduction needs to be significantly improved/revised. While this is extensive, some of the ideas are confusing to the readers and/or even repeated. One example: the sense of “Radiotherapy and chemotherapy are also the most common methods for treating cancer” in lines 47-48 is confusing. Moreover, this is repeated in lines 89-90. 
-    Lines 94-95: this is conclusions. It should be deleted from introduction.
-    There is no discussion in this manuscript. In fact, lines 218- 249 refer to introduction text and/or general concepts; lines 250-271: repetition of results. 
-    Rephrase Figures captions for simplicity. Eg “Fig 2. Effect of PPO on the the migration of A) prostate cancer cell C4-2,  B) lung cancer cell A549, C) breast cancer cell 4T1”- 
-    Improve title Table 1 (This is effect on cell viability /cytotoxic effects … not just IC50 ….) . Also detail the name of the cell lines as footnote.
-    Lines 161-172: do not describe all the IC50 values in the text since these are in the table (there is no need to repeat all. Just highlight the most relevant information).
-    IC50 values in Table 1 are confusing when compared to Fig 1. Eg at 0.5 mg)/ml, values of cell viability of several cell lines are above 50% = MCF-7; GES-1; PC-9; A549 while their IC50 in Table 1 is bellow that value.
-    Fig 1 E: Y axis is not concentration…

Others: 
-    Lines 37/38: delete “Many researchers have studied the treatment of cancer.”
-    Lines 38/39: improve sentence
-    Line 44: some studies
-    Line 49: adriamycin -replace by doxorubin
-    Lines 54- 76: give more details 

Reviewer 2 Report

Please, refer to the attached file. Thank you.

Reviewer 3 Report

Authors:

The idea to investigate polyphenol oxidase as a promissing alternative therapeutic agent for cancer therapy seem to be interesting, and possibly important aim. However, using enzymes to treat cancer in the living organism brings important risk of damaging normal cells as well. After reading the manuscript, it is not clear to me, what the safe activity that can ensure that only cancer cells will be treated, while treatment of the normal cells will be neggligible. This point is not enough explained and discussed.

Moreover, the Table 1 declares IC50 values of cells... . It should be clear enough that the presented IC50 values are not values "OF CELLS", however, values of the enzyme activity "IN THE CELLS".

Figure 1: The differences in viability of the normal cells and the cancer cells seem to be very similar, e.g., in Figure 1b, 1c and 1e.

Figure 2: No data are given for either of the normal cell lines that should be added for comparison of the effect of PPO in the cells, both normal and cancer. I recommend to add the missing data.

Besides this, there are certain numbers of formal insufficiencies throughout the manuscript:

IC50 should be written with the number "50" being in subscript.

Typing errors were found in lines 212, 252, 260 and 261.

References: What does the character [J] appearing after the article names in the references mean? If this is to show that the reference comes from a journal, then [J] seems to be excessive.

I do not recommend to publish this manuscript in its present form.